# Phytochelatins: Sulfur-Containing Metal(loid)-Chelating Ligands in Plants

**DOI:** 10.3390/ijms24032430

**Published:** 2023-01-26

**Authors:** Ilya V. Seregin, Anna D. Kozhevnikova

**Affiliations:** K.A. Timiryazev Institute of Plant Physiology, Russian Academy of Sciences, Botanicheskaya St., 35, 127276 Moscow, Russia

**Keywords:** metal and metalloid accumulation in plants, metal and metalloid detoxification, metal and metalloid transport, phytochelatins, phytochelatin synthase, stress

## Abstract

Phytochelatins (PCs) are small cysteine-rich peptides capable of binding metal(loid)s via SH-groups. Although the biosynthesis of PCs can be induced in vivo by various metal(loid)s, PCs are mainly involved in the detoxification of cadmium and arsenic (III), as well as mercury, zinc, lead, and copper ions, which have high affinities for S-containing ligands. The present review provides a comprehensive account of the recent data on PC biosynthesis, structure, and role in metal(loid) transport and sequestration in the vacuoles of plant cells. A comparative analysis of PC accumulation in hyperaccumulator plants, which accumulate metal(loid)s in their shoots, and in the excluders, which accumulate metal(loid)s in their roots, investigates the question of whether the endogenous PC concentration determines a plant’s tolerance to metal(loid)s. Summarizing the available data, it can be concluded that PCs are not involved in metal(loid) hyperaccumulation machinery, though they play a key role in metal(loid) homeostasis. Unraveling the physiological role of metal(loid)-binding ligands is a fundamental problem of modern molecular biology, plant physiology, ionomics, and toxicology, and is important for the development of technologies used in phytoremediation, biofortification, and phytomining.

## 1. Introduction

Metals and metalloids are natural components of the earth’s crust. Some metals, such as copper (Cu), manganese (Mn), nickel (Ni), zinc (Zn), and iron (Fe), are essential for most living organisms, while the biological roles of cadmium (Cd), mercury (Hg), lead (Pb), and arsenic (As), with rare exceptions, are unknown and they are toxic even at fairly low concentrations in the environment [1,2,3]. When essential elements are supplied in supraoptimal quantities, multiple toxic effects on a large number of physiological processes can be observed, as was shown for Cd, Pb [4,5], Ni [6,7], As [8], Zn [9] and other metals and metalloids [2,3], which is often accompanied by impaired growth and morphogenesis [10]. Due to human activity, the release of metal(loid)s into the environment has increased significantly in recent decades, including contamination resulting from mining, the intensive use of fertilizers, the combustion of liquid and solid fuels, and the development of metal smelting production [1,11]. Unlike organic compounds which can be decomposed by microorganisms, metals and metalloids do not decompose and, therefore, accumulate in the environment and are absorbed by plants, which are the main source of their entry into food chains [12]. The ability of plants to detoxify metals and metalloids, which are naturally accumulated in the soil due to the processes of rock weathering, is an ancient and widespread trait in plants [13]. They use various strategies to detoxify metals and metalloids: sequestration, exclusion, and chelation [3].

In addition to transporters, that mediate metal(loid) translocation across biological membranes [14,15,16,17], a key role in metal(loid) detoxification, transport, and homeostasis belongs to low-molecular-weight ligands capable of forming stable complexes with metal(loid)s. These include sulfur-containing ligands (for example, glutathione and phytochelatins), nitrogen-/oxygen-containing ligands (S-adenosyl-L-methionine derivatives, histidine, and other amino acids), and oxygen-containing ligands (e.g., phenols and organic acids) [18,19]. The metal fraction that can be exchanged between different ligand molecules is termed the labile pool, the maintenance of which is regarded as a key function of low-molecular-weight ligands [18,20,21].

Different plant species, as well as populations, can differ significantly in their ability to accumulate metals and metalloids in roots and shoots. Unlike excluders, that accumulate metal(loid)s primarily in their roots, hyperaccumulators are plant species in which the metal(loid) concentration in their shoots (per gram of dry weight) exceeds 100 μg of Cd, thallium (Tl), or selenium (Se); 300 µg of Cu, cobalt (Co), or chromium (Cr); 1000 µg of Ni, As, Pb, or rare-earth elements; 3000 µg of Zn; or 10,000 µg of Mn under natural growth conditions, which is much higher than in non-accumulating species [22]. Plants from different populations of the hyperaccumulators *Noccaea caerulescens* [23,24,25,26] and *Arabidopsis halleri* [27,28,29] vary in their tolerance to, and capacity to hyperaccumulate, Zn, Cd, Ni, or Zn, Cd, Pb, respectively.

The mechanisms of metal(loid) hyperaccumulation can potentially be controlled at four levels: (1) at the level of metal(loid) uptake from soil by plant root systems, (2) at the level of radial transport of metal(loid)s in roots, (3) at the level of their translocation to the aboveground organs via the xylem, and (4) at the level of accumulation in leaves in a non-toxic form [17,19,30]. Studying the molecular mechanisms of metal(loid) detoxification, as well as the mechanisms that determine the selective accumulation of metal(loid)s in various plant organs, is an important task of modern molecular biology, plant physiology, ionomics, and toxicology.

The phytochelatin-mediated detoxification of metal(loid) ions has been firmly established as a fundamental detoxification mechanism in plants. For the first time, low-molecular-weight, cysteine-rich polypeptides, capable of binding metal(loid) ions via the SH groups of cysteine residues, were found independently in *Schizosaccharomyces pombe* [31] and in *Rauvolfia serpentina* cell cultures [32]. Such peptides were called cadistins A and B [31] or phytochelatins (PCs) [32]. To date, PCs have been found not only in angiosperms and gymnosperms, but also in algae, liverworts, fungi, microorganisms, and some animals [14,33,34,35,36,37], which indicates their appearance at the early stages of evolution [38]. Phytochelatins were found in various foods of plant origin, which makes it important to study their impact on the human body [39].

Based on the available literature data, this review summarizes the modern concepts of various PC families in different plant species, PC biosynthesis, and PC participation in metal(loid) uptake and sequestration in the vacuole, as well as in the long-distance transport of metal(loid)s. In addition, we will try to answer the intriguing question of whether PCs are involved in the mechanisms of hyperaccumulation. For convenience, “metals” and “metalloids” will be referred to as “metals” throughout the text of the review. We apologize in advance to all authors whose papers were not cited in our review due to its limited scope.

## 2. Structure and Accumulation of Phytochelatins in Plants

### 2.1. Metal(loid)-Induced Phytochelatin Production in Plants

The structure of PCs has been determined for a wide range of plant species from different families (Table 1 and Table 2). 

The basic structure of PCs is (γ-Glu-Cys)_n_-Gly, where n = 2–11 but usually does not exceed 4–5 (Table 1) [14,33,110,111,112,113,114,115]. Other possible structures of PCs will be discussed below. The degree of PC polymerization, as well as PC accumulation in plants, largely depends on the physicochemical properties of the metal ions, the duration of exposure, and the metal concentration [44,53,75,76,92,116,117,118,119]. Phytochelatins with a longer chain were usually synthesized only after a lag period [44,75,76,120].

The kinetics of Cd uptake and PC induction during the first 4 h of incubation, and at different levels of N and P, was recently studied on the diatom *Thalassiosira weissflogii* [121] and traced for 14 and 21 days, respectively, in the roots and shoots of *Spinacia oleracea* [42] and *Arabidopsis thaliana* [53]. For example, when *S. oleracea* plants were grown in hydroponics at 3–9 mg/L Cd, glutathione and PC_3_ were predominant on most of the days, and the concentrations of PC_2_, PC_3_ and PC_4_ in the leaves usually reached a peak after 7 or 9 days of exposure, but subsequently decreased during the following days [42]. The concentration of PCs in the roots and leaves of *A. thaliana* plants significantly increased after 3 days of exposure to 5 µM Cd in a nutrient solution, compared to that after 1 day of exposure. After 7, 14, and 21 days of exposure, PC concentration in the roots of Cd-treated *A. thaliana* slightly increased and then remained at a similar level, whereas PC concentration in the leaves reached a peak after 7 days of exposure and subsequently decreased [53]. The kinetics of PC accumulation was also traced for the roots, stems, and leaves of As-treated *Helianthus annuus* [47], roots and leaves of As-treated *Salix atrocinerea* [99], roots of Cd-treated *Pisum sativum* [73] and *Zea mays* [88,91,92,93], roots and shoots of Cd-treated *Secale cereale* and *Triticum vulgare* [88], and gametophytes of Cd-treated *Marchantia polymorpha* [76] (Table 2).

Various metal(loid) ions can induce PC biosynthesis. When different metal(loid)s were added to *R. serpentina* cell cultures at high, but non-lethal, concentrations, PC biosynthesis was induced in the presence of Pb^2+^, Zn^2+^ (1 mM); Cd^2+^, Ni^2+^, tin (Sn^2+^), SeO_3_^2−^, bismuth (Bi^3+^) (100 μM); silver (Ag^+^), Cu^2+^, gold (Au^+^) (50 μM); AsO_4_^3-^ (20 μM); antimony (Sb^3+^) and tellurium (Te^4+^) (10 μM) [44]. Similar results were obtained for cell cultures of *Rubia tinctorum* [98]. Recently, the induction of PC biosynthesis was shown in *Sinapis alba* under treatment with platinum (Pt), rhodium (Rh), and palladium (Pd) [65], and in *Z. mays* seedlings in the presence of vanadium (V) [122]. Although PC biosynthesis can be induced in vivo by various elements, they are mainly involved in the detoxification of Cd and As (III) [14,54,99,123,124,125,126,127] and, to a lesser extent, Hg [57,60,109,128,129], Pb [62,74,130,131,132,133], Zn [54,55,74,133,134], Cu [55,74,119,135,136], and Sb(V) [69], which is partly determined by the stability of the metal complexes with S-containing ligands. Boron (B) [137], magnesium (Mg), calcium (Ca), and sodium (Na) [138] did not induce the biosynthesis of PCs. The possible role of PCs in response to drought, low temperatures, salinity, and other stress factors is also discussed, which, however, requires additional studies [139,140,141]. Since PCs are S-containing compounds, the amount of PCs decreases under sulfur (S) deficiency [142].

Different metals induce the biosynthesis of PCs to a different degree in various species. Ahner and Morel [143], comparing the accumulation of PCs in the algae *T. weissflogii*, *Tetraselmis maculate*, and *Emiliania huxleyi* under the treatment with Cd, Cu, Zn, and Pb, showed that the induction of PC production can be not only metal-specific, but also taxon-dependent. The concentration of PCs in *Emiliania huxleyi* was similar in the presence of Cd and Cu [143], while in *Dunaliella tertiolecta*, PC induction was observed at a higher concentration of Zn compared to Cd [144]. In *Sedum alfredii*, PC production was induced in vivo in the presence of Cd and Pb, but not Zn [145], whereas in *Pfaffia glomerata*, it was induced in the presence of As, but not Hg or Pb [41]. In *Clinopodium vulgare*, the biosynthesis of PCs was induced by Cd, Cu, Zn, and, to a lesser extent, Pb [74], and in *A. thaliana*, it was induced by Cd, Zn, and to a lesser extent, Cu [55,134].

Since under field conditions, especially in polluted areas, plants are often exposed to excessive amounts of more than one metal(loid), synergism or antagonism between the elements in their effect on plants can be observed depending on metal physicochemical properties, as well as environmental factors and biological characteristics of plant species. For example, after 14 days of incubation of *Triticum aestivum* in the presence of Cd or Pb, an increase in PC concentrations was observed, being most prominent in the case of Cd. Under the combined treatment, the metals had a synergistic effect on PC biosynthesis, but an antagonistic effect on the biosynthesis of glutathione [146]. The concentrations of PCs, cysteine, and glutathione in the roots of Cd-treated *Oryza sativa* increased in the presence of silicon (Si) [147], whereas in the roots of Cr-treated plants, an increase in the concentration of PCs, but not glutathione, was observed in the presence of Ca [148]. In the roots and shoots of *Nicotiana tabacum,* Cr (50 µM) did not induce the production of glutathione or PCs, but their concentration increased in Cr-stressed plants when they were supplied with Se (2 µM) and molybdenum (Mo) (1 µM) [149]. The concentration of glutathione and PCs also increased in As-treated *O. sativa* in the presence of Si and TiO_2_ nanoparticles, whereas the nanoparticles alone did not affect the biosynthesis of PCs [150]. Differences between separate and combined effects of metals have also been shown for other ligands [19,151] and, in general, require more thorough studies.

### 2.2. The Structure of Phytochelatin Complexes with Metal(loid)s

Various analytical methods are used to identify PCs and their complexes with metals [38,56,74,83,120,152,153,154], and their advantages and disadvantages are discussed in the review by Ahmad and co-authors [155]. The most numerous studies are devoted to the role of glutathione and PCs in the detoxification of Cd, since Cd ions have the highest affinity for these ligands compared to other metals (Mn, Fe, Cu, Zn) [156]. In spectrophotometric studies, log K^7.4^ values for Cd complexes with ligands (1:1) were 4.8 for glutathione, 6.2 for PC_2_, 7.5 for PC_4_, and 5.5 for PC_6_ [157]. Potentiometric and spectroscopic studies showed that the affinity of Cd complexes increased from glutathione to PC_4_ almost linearly from the micromolar (log *K*^7.4^_GSH_ = 5.93) to the femtomolar range (log *K*^7.4^_PC4_ = 13.39), and additional chain elongation did not significantly increase the stability [120]. The thermodynamic stability of PC complexes with other metals decreased in the following order: Zn^2+^ ≥ Cu^2+^ ≥ Fe^2+^ > Mg^2+^ > Ca^2+^ [158]. Hence, PCs are more efficient for metal chelation than glutathione.

The structure of PC complexes with metal(loid)s can differ both for different metals and for complexes with different degrees of polymerization [38,120,152]. For example, all peptides can form 1:1 Cd-ligand complexes, but 1:2 Cd-ligand complexes were found for glutathione, PC_2_, and, partially, for PC_3_. Moreover, binuclear species, Cd_x_-ligand_y_, were identified for the series PC_3_−PC_6_ under Cd excess [120]. In the leaves of As-tolerant *Holcus lanatus*, the As(III)-PC_3_ complex was the predominant one, although reduced glutathione, PC_2_, and PC_3_ were found in the extract, whereas the As hyperaccumulator *Pteris cretica* only synthesized PC_2_ and formed predominantly the GS-As(III)-PC_2_ complexes [97]. In the roots, stems, and leaves of As-treated *H. annuus*, complexes of arsenite with glutathione, As(III)–PC_3_, and GS–As(III)–PC_2_ complexes were detected. The roots also contained As(III)–(PC_2_)_2_ and monomethylarsonic PC_2_ complexes [47].

PC complexes with metals are stable and less toxic than free metal ions [14,113,114]. For example, PC–Cd complexes are approximately 1000 times less toxic to enzymes than free Cd ions [159]. Therefore, even nanomolar amounts of Cd, which can get into the nutrient solution from contaminated chemicals, can induce PC biosynthesis [160]. However, the amount of PCs as well as the stability of the resulting complexes of PCs with metals depend on the pH [46,161,162]. At the pH of the cytosol (7.2–7.5), the complexes are stable, whereas at the pH of the vacuolar sap (4.5–6.0), they break up [161], which plays an important role in PC-mediated metal detoxification. The formation of stable complexes of PCs with metals in the cytoplasm prevents metal binding to the sulfhydryl groups of proteins, and therefore, the toxic effects on metabolic processes are reduced.

### 2.3. Classification of Phytochelatins and Their Accumulation in Different Plant Organs

The presence of the γ-Glu bond in PCs indicates that they are not primary gene products [44] and therefore are classified as a separate third class of metallothioneins [163,164]. Instead of glycine, at the C-terminus of the polypeptide there can be glutamine (Gln), glutamic acid (Glu), serine (Ser), alanine (Ala), β-alanine (β-Ala), asparagine (Asn), or cysteine (Cys), or the C-terminal amino acid may be absent. In addition, there have been found several PC derivatives that lack either Glu or Cys residues from the γ-Glu-Cys structure [45,83,86,99]. Thus, several families of PCs have been currently identified [83,110,112,165] (Table 1). Some of them are probably specific for some families of angiosperms. Thus, homophytochelatins (homoPCs), with C-terminal β-Ala, have been currently found mainly in legumes [71,72,73,166], and hydroxymethylphytochelatins (hydroxymethylPCs), with C-terminal Ser, have been detected in cereals [82,83,88,89,90,94,116,152]. PCs with C-terminal Glu were found in the roots of *Z. mays* [88,92,93,167], as well as the roots and shoots of *O. sativa* [82,83,152]. In addition, PCs with C-terminal Gln, Asn, and Cys were found in the roots and shoots of *O. sativa* [83]. Phytochelatins that do not contain a C-terminal amino acid, and have the structure γ-Glu(Cys)_n_, have been isolated, for example, from the roots of *Z. mays* [88,92,93], *Capsicum annuum* [118], the roots and shoots of *T. aestivum*, *S. cereale* [88], *O. sativa* [83,152], *Dettrichia viscose* [45], *Panicum maximum* [86], *Betula pubescens* [168], *S. atrocinerea* [99], as well as from the root culture of *R. tinctorum* [98,169] (Table 1 and Table 2). Phytochelatins with C-terminal Ala, Glu, Gln, Ser, Asn, or Cys, and other PC derivatives, have also been proposed to be called iso-phytochelatins (iso-PCs) [83,170].

In the cell culture of *A. thaliana*, both PCs and all types of iso-PCs were found, which indicates the presence of iso-glutathione [58], which acts as an acceptor of γ-Glu-Cys residues during the biosynthesis of PCs [171]. However, in intact plants of *A. thaliana* [48,49,50,51,52,53,54,55,56], as well as in other Brassicaceae species [44,48,59,60,61,62,63], only PCs were found (Table 2). Therefore, the presence of iso-PCs in intact plants from this family remains debatable. Phytochelatins (PC_2–5_) and various iso-PCs were found in *C. vulgare* from the Lamiaceae family [74], whereas homoPCs were found in cereals [86], which suggests a wider distribution of iso-PCs in nature. However, it is still unknown why such a diverse group of PCs is synthesized in various taxa.

In addition to the differences in the terminal amino acid and the degree of polymerization, sulfur in PCs can be present in different oxidation states (thiol versus disulfide form), and metals can theoretically interact with various functional groups of peptides [165]. Since at pH values close to neutral thiol groups are predominantly in the thiolate form (-S^−^), it is assumed that metal binds mainly to them, although the participation of carboxyl groups cannot be excluded [60,120,130,154].

The accumulation of PCs in different plant organs can vary significantly. For example, in *B. pubescens*, *Halimione portulacoides*, and *Sarcocornia perennis* growing on polluted soil [168,172], in *Brassica juncea* under long-term treatment with Cd [173], as well as in Cd-treated *Perilla frutescens* [75], the concentration of PCs in the leaves was higher than in the roots. At the same time, in *Spartina maritima* growing on polluted soil [172], Cd-treated *T. aestivum* [90], Cd- and Zn-treated *A. thaliana* [134], *C. vulgare* exposed to Cd, Cu, Zn, Pb [74], As-treated *Eupatorium cannabinum* and *S. atrocinerea* [46,99], and in Pb-treated *Brassica juncea* [62], a higher level of PCs was observed in the roots. In *O. sativa* seedlings exposed to 50 or 100 µM Cd, a significant variation in the concentrations of various PC derivatives was observed in the roots, stems, and leaves [83], whereas under the exposure to As (V) (100 µM), no significant differences in the concentrations of PCs in the roots and shoots were found [84]. In addition to species-specific features, such differences can be caused by different metal concentrations, duration of exposure, and growth media, which should be taken into account in a comparative analysis of the data obtained by different authors. At the same time, the PC concentration did not always correlate with the metal(-loid) concentration, which is a reflection of the existence of other metal detoxification mechanisms that can be acting simultaneously and with different efficiency in different species, such as, for example, binding to the material of cell walls [174], metallothioneins [35,37,175], or other ligands [19]. Significant differences in the concentration of PCs may also be a reflection of plant cultivar characteristics, which, for example, was shown in a comparative analysis of the accumulation of PCs and iso-PCs in the roots of plants of twelve *O. sativum* cultivars exposed to As [152] and two *Brassica parachinensis* cultivars exposed to Cd [176].

## 3. Biosynthesis of Phytochelatins and Its Regulation

### 3.1. Glutathione as a Precursor of Phytochelatins

Biosynthesis of PCs requires L-glutamate (Glu), L-cysteine (Cys), and glycine (Gly). Metals affect various stages of PC biosynthesis, from S assimilation to glutathione biosynthesis [37,177]. PCs are synthesized from reduced glutathione, which is one of the reasons for the decrease in the pool of intracellular glutathione [44,177]. A Cd-induced decrease in the concentration of glutathione was observed, for example, in *Pontederia* (*Eichhornia*) *crassipes* [96], *Arachis hypogaea* [70], *P. sativum* seedlings [72], in the roots of *Pistia stratiotes* [178], *O. sativa* [83,85], and *Z. mays* [88,179,180], and in suspension-cultured cells of *Solanum lycopersicum* [102], while an As-induced decrease in the concentration of glutathione was shown in the roots of *S. atrocinerea* [99] and *O. sativa* [84]. However, in the hyperaccumulators *A*. *halleri, N*. *caerulescens*, and *S. alfredii* [48,181], the accumulator *P. frutescens* [75], in the roots of *Phragmites australis* [87] and *Cajanus cajan* [182], in the roots and shoots of *Solanum lycopersicum* [108], and in the liverwort *M. polymorpha* [76,77], in the presence of Cd an opposite pattern was observed. The effect of Cd, as well as other metals, on the content of glutathione may differ depending on the plant organ, the duration of exposure, and the concentration of the metal in the medium [42], which may partly explain the conflicting results obtained for *B. juncea* [61,183,184] and *A. thaliana* [48,55]. In addition, the metal’s effect may depend on the endogenous content of glutathione. Thus, the glutathione concentration in *M. polymorpha* gametophytes was constitutively low compared to higher plants, but a significant increase in its level was observed under the treatment with Cd [76,77].

The biosynthesis of glutathione from Glu, Cys, and Gly is an ATP-dependent, two-step process (Figure 1) [177,185,186]. The first reaction for the formation of γ-Glu-Cys from Glu and Cys takes place in the chloroplasts and is catalyzed by glutamate cysteine ligase (EC 6.3.2.2), previously known as γ-glutamylcysteine synthase [177,186,187,188,189], which is encoded by the *GSH1* (*GCS*) gene [190,191] (Figure 1). The reaction catalyzed by this enzyme is considered as the rate-limiting step in the biosynthesis of glutathione [103,126,185,186,189,191]. The second stage of the biosynthesis of glutathione from γ-Glu-Cys and Gly can occur both in the chloroplasts and in the cytosol, and is catalyzed by glutathione synthetase (EC 6.3.2.3), which is encoded by the *GSH2* (*GS*) gene (Figure 1) [177,186,189,192]. Glutathione that is synthesized in the chloroplasts enters the cytosol, where it is involved in the biosynthesis of PCs, via the chloroquine resistance-like transporters (CLT1–3) (Figure 1) [188].

The biosynthesis of γ-glutamylcysteine from L-cysteine and L-glutamate (Glu) (stage 1) takes place in chloroplasts and is catalyzed by glutamate cysteine ligase encoded by the *GSH1* gene. The formation of glutathione (GSH) from γ-glutamylcysteine and L-glycine (Gly) (stage 2) can occur both in the chloroplasts and in the cytosol (the latter is not shown in the scheme) and is catalyzed by glutathione synthetase encoded by the *GSH2* gene. Glutathione is transported from the chloroplast to the cytosol via the CLT1–3 transporters. The biosynthesis of phytochelatins (PCs) with the participation of phytochelatin synthase (PCS), the activity of which increases (+) in the presence of metal ions (Me^n+^), takes place in the cytosol (stage 3). Phytochelatin synthase is encoded by the *PCS* genes, whose expression can change under plant exposure to metals. In some plant species, iso-phytochelatins were found, which are synthesized with the participation of iso-GSHs. In the cytosol, metal ions bind to phytochelatins, with the formation of various complexes that differ in the structure and degree of polymerization of phytochelatins. The ATP-dependent transport of low-molecular-weight metal complexes with phytochelatins (LMW Me-PC complexes) across the tonoplast is carried out by ABCC1/2/3 transporters. In the vacuole, high-molecular-weight complexes of metals with phytochelatins (HMW Me-PC complexes) can be formed, with the participation of acid-labile sulfide (S^2-^) presumably coming from chloroplasts. Due to the acidic pH values of the vacuolar sap, it is likely that these complexes can be partially destroyed, and metal ions can bind to organic acids (OAs), forming, for example, citrates and malates. The proposed processes are indicated by dotted lines.

Glutathione is present in almost all cell compartments. It is a strong reducing agent and is easily oxidized, participating in many processes, including metal binding, inactivation of reactive oxygen species, and regulation of redox homeostatic processes [33,34,53,123,126,177,185,186,193,194]. Glutathione exists in reduced and oxidized form. Glutathione reductase catalyzes the conversion of the oxidized form into the reduced one [186,192]. The ratio between these two forms is an indicator of redox balance and its maintenance at a certain level is crucial for plant survival [33]. Cadmium-induced decrease in the concentration of glutathione was accompanied by an increase in the activity of glutamate cysteine ligase, glutathione synthetase [195,196,197,198], glutathione transferase [199,200], and ATP-sulphurylase [183,201,202]. Thus, the induction of PC biosynthesis by metals can be achieved due to an increase in the activity of glutamate cysteine ligase and glutathione synthetase, involved in the biosynthesis of glutathione [203]. The increase in the activity of glutathione biosynthesis enzymes correlated with an increase in the expression of the *GSH1* and *GSH2* genes [99,108,150,192,204,205]. However, along with an increase in the concentration of thiol-peptides under the action of Cd, Cu, and Zn, no stimulation of the expression of the *AtGSH1* and *AtGSH2* genes was found in *A. thaliana*, which, as suggested, points to the existence of only post-transcriptional control [55]. Similar results were obtained, for example, when studying the effect of Cd on *PmGSH1* expression in *P. maximum* [86]. At a high concentration of glutathione, its biosynthesis is regulated by a negative feedback, as well as by other mechanisms [177].

The mutants of *S. pombe* and *A. thaliana* with a reduced concentration of glutathione were characterized by PC deficiency and hypersensitivity to Cd [113,206,207,208], Hg, and As [209]. Treatment of *Glycine max* with L-buthionine-sulfoximine (BSO), an inhibitor of glutathione biosynthesis, led to a decrease in the PC accumulation after 5 days of incubation in the presence of As (III) and As (V), and an increase in As translocation into the leaves [194]. Transgenic *B. juncea* plants, overexpressing the *GSH1* and *GSH2* genes, contained more glutathione and PCs, were more tolerant to Cd, and accumulated more Cd than wild type plants [210,211]. However, not in all studied plant species did an increase in the level of glutathione lead to an increase in metal tolerance [177]. Nevertheless, the intracellular level of glutathione is one of the important regulators of PC biosynthesis, and the activation of glutathione biosynthesis in the presence of Cd can also occur as a result of the metal-induced production of reactive oxygen species [192]. Therefore, metal tolerance of the excluder species is determined not only by PC biosynthesis, but also by the ability to maintain the intracellular concentration of glutathione at an optimal level.

### 3.2. Phytochelatin Synthase Is a Key Enzyme in the Biosynthesis of Phytochelatins

PC biosynthesis from reduced glutathione is catalyzed by the key enzyme γ-glutamylcysteine dipeptidyl (trans)peptidase (phytochelatin synthase, PCS) (EC 2.3.2.15), belonging to clan CA of the papain-like cysteine proteases [14,33,212,213], whose activity is regulated at the transcriptional and post-transcriptional levels [112,214]. The enzyme is around 95,000-Mr tetramer, with a Km of 6.7 mM for glutathione [112]. The biosynthesis of PCS seems to be constitutive in plant cells and cell cultures [99,212]. Even at a low metal concentration, PCSs in prokaryotes and eukaryotes are able to provide the basic level of PCs in the cell, which indicates their role in maintaining ion homeostasis and regulating the availability of metal ions in the cell [46,76,215,216]. The reaction catalyzed by this enzyme is an autotransferase reaction, in which up to 10 dipeptidyl residues can be transferred, as was demonstrated for plant cells [72,105,124,217].

Phytochelatin synthases are evolutionarily conserved in different species of higher plants and charophytes [218] and consist of 452–545 amino acid residues with a characteristic but variable C-terminal domain called the Phytochelatin_C domain [112,219,220]. It contains numerous Cys residues involved in metal binding and determining the increased stability of the protein as well as broad metal specificity [36,52,57,138,219,220,221,222,223]. Different regions of the C-terminal domain of AtPCS1 in *A. thaliana* are important for its activation by Cd, Hg, Zn, and As (III) ions [54,219,221,224]. At the same time, the mechanism of AtPCS1 activation under the action of phenylmercury (PheHg) and Hg may be similar [57]. Deletion of the last 10 amino acid residues of the C-terminal domain of AtPCS1 led to an increase in As(III)-dependent PC biosynthesis, which indicates the involvement of some amino acid residues in this region in the inhibition of PCS activation by As ions [221]. The existence of a Cd- and Zn-dependent mechanism of enzyme inhibition to prevent its overactivation has recently been revealed. It involves two twin-Cys motifs in the C-terminus of MpPCS in *M. polymorpha* [36]. Phylogenetic analysis has shown that the N-terminal domain of PCS, known as the Phytochelatin domain [112,219], is more conservative and has catalytic activity, in which Cys-56, His-162, and Asp-180 residues play an important role [36,52,212,219,220,223,224,225,226]. The position of these specific amino acid residues may slightly vary between PCSs from different plant species. For example, the catalytic triad in BnPCS1 from *Boehmeria nivea* is represented by the residues Cys-58, His-164, and Asp-182 [203]. In *A. thaliana*, Glu-52 in AtPCS1 plays an important role in providing plant tolerance to As and Sb [227].

Despite the fact that PC biosynthesis occurs in the cytosol, PCS has also been found in some organelles. It has been shown that AtPCS1 is localized in the cytosol of root and shoot cells [228,229]. However, its distribution in plant organs is tissue-specific. Cell-type specific expression of *AtPCS1-GFP* in the roots of *cad1-3/pAtPCS1-AtPCS1cds-GFP* line was detected in the rhizodermal cells in the mature and elongation zones, and the outer-most layer of the lateral root cap cells in the meristematic zone [57]. In the shoots of *A. thaliana*, *AtPCS1-eGFP* expression was found mainly in the epidermal cells [228,229], while in *Vicia sativa,* VsPCS was localized in the cytoplasm of mesophyll protoplasts [52]. It has been shown that ZmPCS1 from *Z. mays* is also a cytoplasm-localized protein [230]. The revealed patterns may be related to the fact that rhizodermal cells are the first to contact with metals, while in shoots, metals often accumulate in the epidermal cells, which results in a decrease in their concentration in the mesophyll, and, consequently, in the manifestation of their toxic effects [19,174]. In *S. pombe*, SpPCS1 was localized in mitochondria [231], in *Saccharum officinarum* SoPCS was found in the cytosol and mitochondria [232], in *O. sativum* OsPCS1 and OsPCS2 were localized in the cytosol [233], whereas BnPCS1 in *B. nivea* [203] and AtPCS2 in *A. thaliana* [234] were found in the cytoplasm and nucleus. The data obtained confirm the possible involvement of PCS in the maintenance of ion homeostasis not only in the cytosol, but also in various organelles. They also indirectly indicate the participation of this enzyme in various processes in the cell.

Most plant PCSs are of plastid origin, since they are functionally similar to cyanobacterial PCSs, which, however, do not contain a C-terminal domain [212,235]. For example, a comparative analysis of PCSs from the gametophytes of *M. polymorpha* and the cyanobacterium *Geitlerinema* sp. strain PCC 7407 showed a similar pronounced transpeptidase activity under the action of Cd [76]. It is hypothesized that the mature full-length PCS in higher plants may have evolved from the cyanobacterial protein by the acquisition of more Cys residues in the N-terminal domain and by fusion with a C-terminal domain either from their own genomes or from that of another species [235]. From an evolutionary point of view, these data justify the opinion that the high metal concentration in the environment at the dawn of life could have contributed to the appearance of a metal detoxification mechanism involving PCSs in some ancient groups of organisms [236].

The molecular mechanism of PC biosynthesis was first proposed in 1989 by Grill et al. [217], who suggested that at the first stage, prior to the transpeptidase reaction, Gly is cleaved from glutathione. Then, at the second step, the remaining γ-Glu-Cys forms a peptide bond either with glutathione to form PC_2_ or with another PC molecule acting as an acceptor, resulting in the formation of the PC_n+1_ oligomer [217]. It was later confirmed that AtPCS1 is a dipeptidyltransferase, which undergoes γ-Glu-Cys acylation at two sites, with the release of Gly, during step I of the catalysis that is necessary for net PC synthesis, but the requirements for each acylation reaction are distinguishable. Kinetic studies have shown that one of the substrate binding sites has a high affinity, and the other has a low affinity, for glutathione [237]. In a medium lacking Cd ions, acylation of glutathione occurs at the first site [212,237], which contains a sequence conserved in all PCSs, including Cys, His, and Asp residues, and is metal-independent [238]. Acylation at the second site occurs only in the presence of metal ions, resulting in metal-dependent catalysis [212,238]. The proposed model of the work of eukaryotic PCS suggests that the residues Cys-56, His-162, and Asp-180 in the N-terminal domain are required for the catalytic process, and a certain sequence of the C-terminal domain is responsible for the interaction with the Metal-GS_2_ complex for the amplification of the catalytic process. The initial PCS reaction step (step I) at site I, when one free glutathione molecule is taken as a substrate, releasing Gly and acylating the enzyme with the remaining γ-EC residue, is catalyzed by residues localized in the N terminal domain and is metal-independent. The second reaction is supposedly carried out in site-II and is metal-dependent. This second step involves the transfer of a γ-EC residue to a second glutathione molecule (to form the Metal-GS_2_ complex), or PC_n_, resulting in a net synthesis of a PC_n+1_ molecule [38,212,222,235,238].

To the greatest extent, PCS is activated by Cd ions via metal binding to a specific activation site [112,225,239,240]. For other metal ions, their activating effect in vitro decreased in the following order: Cd^2+^ > Ag^+^ > Bi^3+^ > Pb^2+^ > Zn^2+^ > Cu^2+^ > Hg^2+^ > Au^+^ [112,124,217]. However, the ability of different metals to activate PCS remains an object for studies, and the results obtained are not always unambiguous. Thus, Vatamaniuk et al. [237] showed that AtPCS1 is capable of synthesizing PCs in the presence of Cu^2+^, Zn^2+^, Mg^2+^, Ni^2+^, or Co^2+^, whereas Oven et al. [171] did not observe the activity of AtPCS1 and GmhPCS1 after the treatment with Mg^2+^, Ni^2+^, or Co^2+^. Purified AtPCS1 and LjPCS1 were activated, in decreasing order, by Cd^2+^, Zn^2+^, Cu^2+^, and Fe^3+^, but not by Co^2+^ or Ni^2+^, in the presence of 5 mM glutathione and 50 mM metal ions [241]. Mo, Co, and Ni ions did not activate OsPCSs [223]. One of the reasons for such differences in the experiments with intact plants may be the heterogeneous distribution and accumulation of metals in plant tissues [174], which determines the different availability of metals for PCS [177]. In addition, PCS paralogs characterized in different species displayed functional differentiation in terms of the amount of PCs produced and the specificity of metal-mediated activation, as well as differential regulation of transcription in response to metals [82,215,242,243].

The effect of metals on the activity of PCS can differ not only between species, but also between different isoforms of the enzyme within species. For example, in *Oryza sativa*, OsPCS1 was activated to a greater extent by As^3+^ than by Cd^2+^, while for OsPCS2, the pattern was the opposite [82,233], and the replacement of even one amino acid residue in the C-terminal domain can affect the metal selectivity of the enzyme [223]. The involvement of various isoforms of PCS in the response to metals requires more thorough study. For example, it was shown that, in contrast to AtPCS1, AtPCS2 was not stimulated by Cd ions, leading to the assumption that only AtPCS1 determines the synthesis of PCs, metal tolerance, and plant ability to accumulate metals [215,244]. However, the results of other studies indicate that AtPCS2 may be involved in the response of plants to Cd during long-term incubation at a higher metal concentration [198]. It is also assumed that the activation of PCS may be mediated by H_2_O_2_, which is formed as a result of metal-induced oxidative stress [115,245], which in turn is consistent with the data on the possible involvement of PCs in the neutralization of H_2_O_2_ and superoxide radicals [35,235]. In addition, PCS activity can be regulated by phosphorylation and dephosphorylation of the enzyme. In vitro experiments demonstrated that PCS activity increased after its phosphorylation by casein kinase 2 (CK2) and decreased after treatment with alkaline phosphatase. Site-directed mutagenesis experiments on AtPCS1 indicate that Thr-49 near the catalytic site in the N-terminal domain is the site for phosphorylation [246]. Thus, the activity of PCS and the amount of available glutathione can be considered as important mechanisms for the regulation of PC synthesis.

The biosynthesis of iso-PCs has not been sufficiently studied yet [247]. It is possible that PCs with C-terminal Ser or Glu are synthesized through ATP-dependent ligation from γ-EC and Ser or Glu, like in glutathione biosynthesis, or through post-synthetic modifications of glutathione, like in the transpeptidation during the PC biosynthesis. It is assumed that in *O. sativa* seedlings, OsPCS2 can catalyze the conversion of glutathione to γ-EC in the cytosol under the action of Cd, after which γ-EC can be used as a substrate for the subsequent synthesis of hydroxymethyl-glutathione or γ-Glu-Cys-Glu [82]. Homoglutathione can be a substrate in the biosynthesis of homoPCs [72,171,241]. However, homoglutathione was found only in the leaf blades of *P. maximum*, while homoPCs were also found in the stems and roots [86]. It was assumed that the biosynthesis of homoPCs was carried out with the participation of homophytochelatin synthase [72,171], while later it was shown that homoPCs can be synthesized with the participation of typical PCS [241]. Therefore, the biosynthesis and transport of iso-PCs require further research. The presence in many plant species of PCs that do not contain a C-terminal amino acid (Table 2) is the result of its cleavage from PCs, homoPCs, and hydroxymethylPCs by peptidase, or due to the hydrolytic activity of PCS [72].

### 3.3. Phytochelatin Synthase Genes in Living Organisms: Identification and Expression

Genes encoding PCSs have been found in species from distant taxa, including bacteria [235], fungi [124,231,248], Metazoa (ciliates, flatworms and annelids, echinoderms, chordates) [124,249,250,251], algae [236], and higher plants. Over the past two decades, from the pioneering works [225,248,252], numerous *PCS* gene orthologues have been identified and characterized. Many plant species have more than one copy of functionally active *PCS* genes. *PCS* genes have been identified in a number of plant species (Table 3). 

*PCS* genes have also been identified in the yeast *Schizosaccharomyces pombe* (*SpPCS1*) [225,231], the nematodes *Caenorhabditis elegans* (*CePSC1*) [249] and *Ancylostoma ceylanicum* (*AcePCS*) [250], the tunicate *Ciona intestinalis* (*CiPCS*) [251], and other species [38], which has made it possible to carry out phylogenetic analyses [36,52,124,213,226,232,236,255,270].

From an evolutionary point of view, the absence of the *PCS* gene in the genome of the model moss *Physcomitrium patens* is of interest, which indicates the leading role of other mechanisms of metal detoxification in this species [281] and is consistent with a very low level of PCs in the Cd-tolerant moss *Leptodictyum riparium* [200]. However, in the liverwort *M. polymorpha*, not only the presence of one copy of the *PCS* gene was shown [36], but also the participation of MpPCS in the detoxification of Cd, but not As (III) or other divalent cations [12]. It is assumed that *PCS* genes probably have a bacterial origin and were subsequently inherited to different groups of organisms, in some cases multiple times. It was suggested that multiple horizontal gene transfer events from bacteria to eukaryotes occurred within the *PCS* gene family. The complex evolution of the *PCS* genes involves several gene duplications and losses, or independent insertions of the full-length *PCS* genes, in plants and green algae [236].

The presence of several *cis*-regulatory elements in the promoter regions of *PCS* genes, including stress-responsive elements, may explain the influence of metals and other stress factors on *PCS* gene expression [220]. MYB40 transcription factor was shown to regulate the expression of *PCS.* In *A. thaliana* plants, treatment with As(V) induced the expression of *AtMYB40*, which led to the increased expression of *AtPCS1* [282]. The level of expression of *PCS* genes can differ in plant organs and change differently in metal-treated plants. In *A. thaliana* [260], *Lactuca sativa* [257], *B. juncea* [173], *B. parachinensis* [176], *C. cajan* [182], *Salicornia europaea* [141], and *Tagetes patula* [259], the *PCS* gene expression was higher in the roots than in the shoots, while in *H. annuus* [256] the opposite pattern was observed. For *O. sativa*, a higher level of *OsPCS* expression was shown in the roots [85], but the level of *OsPCS1* and *OsPCS2* expression in different organs can vary significantly [82,270], which can also be partly determined by the use of different varieties and plant growth conditions. The expression level of *SoPCS* changed differently in the roots and shoots of *S. officinarum* with an increase in the concentration of Cd in the medium [274]. The Cd-induced increase in the amount of *BnPCS1* mRNA in the leaves of *B. nivea* was significantly higher than in the roots and stems [203], which is consistent with the data for *NnPCS1* [267]. In Cd-treated *Paspalum vaginatum*, the expression of *PvPCS1* and *PvPCS2* in the leaves decreased within 6 h and was up-regulated after 24 h of exposure. In the roots, *PvPCS1* expression showed significant up-regulation after 6 h of treatment, whereas the expression of *PvPCS2* decreased after 6 h of Cd treatment and then returned to control levels [272]. In *Medicago sativa*, the expression level of *MsPCS* increased with Ni concentration to a greater extent in the roots than in the shoots [283]. An increase in the *SlPCS* and *OsPCS* expression was observed mainly in the roots of As-treated *S. lycopersicum* [205] and *O. sativa* [150]. Therefore, the differences in the *PCS* gene expression may be associated with the physiological characteristics of plant species, which determine their different ability to accumulate metals in different organs, as well as with different durations of exposure and metal concentrations tested, and plant varietal characteristics, which, for example, was shown in the analysis of the expression of the *AtPCS1* and *AtPCS2* genes [198,244]. The effect of a variable-valence metal on *PCS* gene expression may depend on its valence. For example, six putative *PCS* genes were expressed differentially in *O. sativa* seedlings exposed to Cr(VI) or Cr(III) [284]. In addition, the level of expression of different *PCS* genes in one organ can also vary, which was clearly shown by a comparative analysis of the expression of three *AdPCS1-3* genes in *Arundo donax* roots under the treatment with Cd [213].

The expression of different *PCS* genes within plant species can be induced by different metals. In *O. sativa*, the expression of *OsPCS7* was induced by Hg and Pb, the expression of *OsPCS9* was induced by Cd and Zn [269], and the expression of *OsPCS5/-15* was induced by Cd and As [271]. The studies of Cd, Cu, Zn, and Ni effects on different *Azolla* species revealed that *PCS1* gene expression was species- and metal-specific, and the expression level depended on both the duration of exposure and metal concentration in the medium [285]. The relative expression of the *MnPCS1* and *MnPCS2* genes increased in the roots, stems, and leaves of *Morus notabilis* after 24 h of incubation, being significantly stronger under the action of Cd than Zn [243]. In the leaves of *S. lycopersicum*, the expression of the *SlPCS1* gene was induced to a greater extent by Cd and Pb compared to Cu [280], while the induction of the *MhPCS* gene expression in *M. hupehensis* decreased in the series Cd ˃ Cu ˃ Pb [276], which, however, is consistent with a significant increase in *PCS* expression in the presence of Cd in other plant species [203,274]. Therefore, despite the constitutive expression of *PCS* genes, the level of expression of these genes is usually higher in the presence of Cd compared to other metals.

The expression level of *PCS* genes is not only metal- and organ-specific, but also depends on the level of S in the medium. The use of Na_2_SO_4_ as an additional source of S led to an increase in the level of *OsPCS* expression and of PC content in the roots of *O. sativa* [85]. Later, it was shown that under the combined treatment with Cd and S, the expression level of not only *MsPCS1*, but also *MsGS* in the roots of *M. sativa* increased, which was accompanied by an increase in the concentrations of glutathione and PCs [266]. The stimulating effect of S on the accumulation of glutathione and PCs was also shown in Cd-treated *Fagopyrum tararicum* [286] and in the roots of Pb-treated *T. aestivum* [275], which confirms the important role of S in metal detoxification. However, in *P. maximum*, no change in the expression of the *PmGSH1* and *PmPCS2* genes in leaves was found under the combined treatment with Cd and S, which was accompanied by multidirectional changes in the content of PCs in the roots and shoots compared with the Cd treatment [86]. The data obtained confirm that further research in this direction is necessary.

Different species of arbuscular mycorrhizal fungi can influence PC production in plants in response to Cd. For example, mycorrhizal inoculations significantly promoted the expression of the *CcPCS1X1*, *CcPCS1X2*, and *CcPCS1X4* genes, more in the roots than in the leaves of *C. cajan*, indicating that symbiosis with arbuscular mycorrhizal fungal species could enhance Cd tolerance by modulating the expression of *PCS* genes in plants [182].

In most of the species studied, the increase in the level of *PCS* expression led to an increase in plant tolerance to Cd [50,52,203,213,230,243,252,254,261,268,273,277,287,288]. However, this effect was dependent on the concentration of Cd in the medium [50]. A significant increase in Cd tolerance in transgenic *A. thaliana* was observed with the overexpression of the *BnPCS1* gene from *B. nivea* [203], the *BnPCS* gene from *Brassica napus* [261], the *NnPCS1* gene from *Nelumbo nucifera* [267], the *VsPCS1* gene from *V. sativa* [52], or the *ZmPCS1* gene from *Z. mays* [230]. The elevated expression of *CdPCS1* from *Cynodon dactylon* [268], *AtPCS1* from *A. thaliana* [50,287], or *PtPCS* from *Populus tomentosa* [277] also increased metal tolerance of transgenic *N. tabacum* plants compared to the wild type plants. Transgenic *N. tabacum* lines overexpressing the *NtPCS1* gene in the sense or antisense direction were characterized by increased tolerance to Cd and As [279]. Overexpression of *MnPCS1* and *MnPCS2* from *M. notabilis* in *A. thaliana* and *N. tabacum* enhanced the tolerance in most transgenic plants not only to Cd, but also to Zn [243]. Transgenic lines of *B. juncea* with an average level of *AtPCS* expression showed increased tolerance to Cd and As [288]. In contrast, the *cad1* mutant of *A. thaliana*, unable to synthesize PCS, was hypersensitive to Cd [49,225,248], and its seeds did not germinate at a high Cd concentration in the medium [51], whereas the *cad1-3* mutant, the *AtPCS1* null mutant lacking a functional PCS1 [49], was very sensitive to Cd and As and, to a lesser extent, to Zn, Pb, Ag, Cu, and Hg [54,131,134,225,240]. The *OsPCS1* mutants of *O. sativa* were also sensitive to Cd and As [240]. In *cad1-3* transgenic mutants expressing the *TaPCS1* [51] and *MpPCS* [36] genes, restoration of PC formation and an increase in Cd tolerance were observed. Moreover, ectopic expression of *ZmPCS1* repaired the defective phenotypes in the Cd-sensitive yeast mutant *Δycf1* and *A. thaliana* AtPCS1-deficient mutant *atpcs1* under Cd stress, enhancing their Cd tolerance [230].

However, a number of studies have shown a decrease in the tolerance of transgenic plants to Cd and Zn [36,207,213,271,289,290]. For example, an increased level of *AtPCS1* expression in *A. thaliana* and *N. tabacum* led to a decrease in Cd tolerance [207,289,290]. Similar results were obtained on transgenic *A. thaliana* plants with overexpression of the *OsPCS5/-15* genes [271]. Several explanations for such discrepancies have been proposed: the supraoptimal content of PCs in relation to glutathione [207], the manifestation of metal-induced oxidative stress in transgenic plants [289], as well as experimental differences, such as the differences in the use of vectors and constructs [230]. Later, it was shown that Cd-hypersensitive *N. tabacum* plants expressing the *AtPCS1* gene from *A. thaliana* had a high activity of PCS, but a significant decrease in the content of glutathione and in the cytosolic and vacuolar PC pools, whereas in Cd-tolerant *N. tabacum* plants expressing the *CePCS* gene from *C. elegans*, no dramatic change in the glutathione content was observed, and the PC content was significantly higher [290]. It is also assumed that with the increased expression of *PCS*, the rate of formation of PC complexes with Cd may exceed the capacity to transport them into the vacuole, as a result of which they are accumulated in the cytosol. This in turn can be a trigger for PC degradation, including due to the peptidase activity of PCS, which leads to an increase in the Cd^2+^ concentration in the cytosol and an increase in the toxic effect [12,290]. Interestingly, when three recently diverged PCS genes (*AdPCS1-3*) from *A. donax* were overexpressed in *A. thaliana* Col-0 wild type plants, it resulted in either enhanced (*AdPCS2* and *AdPCS3*) or decreased (*AdPCS1*) sensitivity to Cd^2+^ [213]. It is obvious that different activity of PCS, different endogenous levels of PCs/glutathione and degrees of PC polymerization, as well as different efficiency of the translocation of metal complexes with PCs into the vacuole in transgenic plants [50,289,290,291], can significantly affect their metal tolerance. Taken together, the data obtained indicate the presence of a very fine regulation of the PC-dependent mechanism of metal detoxification in different plant species, which undoubtedly should be taken into account when creating transgenic plants.

### 3.4. Other Functions of Phytochelatin Synthase

The constitutive expression of *PCS* [292] and the presence of homologues of the *PCS* gene(s) in plants growing in ecosystems geographically remote from metal-contaminated sites, as well as in representatives of various kingdoms of living organisms, suggest that PCS has a wide range of different functions [124]. In addition to a response to metal-induced stress, PCS is a cysteine peptidase that regulates the catabolism of glutathione and glutathione conjugates in the cytosol [228,242,293]. As a result, the glycine residue is cleaved off from the conjugates, similarly to how it occurs during the biosynthesis of PCs. Phytochelatin synthase is also involved in maintaining Fe homeostasis in charophytes [218], *AtPCS1* is involved in the control of pathogen-induced callose deposition [229,291,294], and *AtPCS2* is involved in response to salinity [295]. As to the latter, it is worth mentioning that an increase in the activity of PCS, the expression of the *SePCS1* gene, and the concentration of PCs was shown for the salt-tolerant halophyte species *S. europaea* in response to combined and separate Cd and NaCl treatments [141].

### 3.5. Hormonal Regulation of the Biosynthesis of Phytochelatins

There is a limited number of studies on the hormonal regulation of PC biosynthesis, which are discussed in detail in a review by Pál et al. [296]. In plants, there is hormonal regulation of glutathione biosynthesis, which, as a result, affects the biosynthesis of PCs. No direct relationship was found between the levels of glutathione/PCs and auxins [55]. However, after the treatment with an auxin inhibitor, a decrease in the content of cysteine, glutathione, and PCs was observed in the roots of *O. sativa* [297]. Mutant and transgenic plants of *A. thaliana* and *N. tabacum* with reduced endogenous levels of cytokinins had higher levels of glutathione and PCs, as well as a higher tolerance to As compared to the wild type [298]. Ethylene was shown to induce the expression of the genes involved in the biosynthesis of glutathione [296]. The expression of *BnPCS1* in *B. nivea* [203] and *StPCS1* in the roots of *Solanum tuberosum* [299,300] was induced by exogenous abscisic acid, but not by salicylic acid. However, there are very few direct studies on the effects of ethylene, abscisic and jasmonic acids, as well as gibberellins on PC biosynthesis, which is a promising direction for future research.

## 4. Transport and Physiological Role of Phytochelatins

### 4.1. Phytochelatins in Hyperaccumulators and Excluders

It is generally accepted that glutathione and PCs are involved in the mechanisms of metal detoxification and transport, but not in the mechanisms of metal hyperaccumulation [14,68,113,301,302,303,304]. In the shoots of the hyperaccumulators *A. halleri*, *S. alfredii*, and *N. caerulescens*, a low concentration of PCs was observed or they were completely absent [48,181,305,306]. Interestingly, when *N. caerulescens* plants were grown in hydroponics in the presence of Cd (5–500 μM), PC biosynthesis was induced both in roots and shoots, whereas in plants growing in their natural habitat at an old Cd/Pb/Zn mining and smelter site in Plombières (Belgium), the PCs were practically not detected [64]. In the roots and shoots of *Dianthus carthusianorum* plants from a non-metalliferous soil, a higher level of PCs was found in response to Cd compared to the plants from a metalliferous soil [307]. A similar phenomenon was found in *S. alfredii* [181], *Silene vulgaris* [66], and *D. viscose* (at 5 mg/L Cd) [45]. The treatment with BSO almost completely arrested the biosynthesis of PCs, but did not enhance the sensitivity to Cd in *N. caerulescens* or in *D. carthusianorum* plants from metallicolous populations [64,307]. Consequently, the high tolerance of hyperaccumulators and metallophytes to Cd is not associated with the increased biosynthesis of PCs [63,64,302,307,308]. It was assumed that hypertolerance may be partly determined by a constitutively high concentration of glutathione in hyperaccumulator plants [193,309]. However, in some cases, the differences in the concentration of glutathione between the plants from non-metalliferous and metalliferous soils were not detected or were ambiguous [45,307].

In general, the amount of Cd bound to S-containing ligands in the shoots of hyperaccumulators is quite low, as shown, for example, for *N. caerulescens* [305,310,311], *Noccaea praecox* [312], and *A. halleri* [306,313]. Zinc did not induce the biosynthesis of PCs in *N. caerulescens*, whereas the concentration of PCs increased with the concentration of Cd, but decreased with an increase in the duration of incubation [305]. In the leaves of *S. alfredii*, only 5% of the total amount of Cd was bound to PCs, which, however, does not eliminate their participation in Cd detoxification [68,145]. Although the amount of Cd complexes with S-containing ligands may depend on the duration of exposure [310], a significant amount of Cd in hyperaccumulators is often bound to O-containing ligands, possibly organic acids [306,310,311,313,314]. For Zn and Ni, histidine and nicotianamine can play a leading role in metal binding [19], while for the elements with variable valencies, the situation can be more complicated. For example, the ratio between Cu complexes with S- and O-containing ligands may not only be species- and organ-specific, but may also differ for complexes with Cu(I) and Cu(II), as was shown for the Cu accumulators *Persicaria capitata*, *Persicaria puncata*, and *Conyza cordata* [315].

In contrast to the hyperaccumulators, in the non-accumulator *Arabidopsis lyrata*, the highest amount of Cd was bound to S-containing ligands. This confirms the involvement of glutathione and PCs in Cd detoxification in non-tolerant plant species [313]. Since the post-translational activation of PCS depends on the availability of metal ions or their complexes with glutathione [171], the more efficient PC biosynthesis in the roots of excluders may be associated not only with a higher level of expression of the *PCS* gene, but also with higher availability of metal ions in the roots of these species as compared with hyperaccumulators [48]. Due to the highly efficient functioning of PC-independent metal detoxification pathways in the shoots of hyperaccumulators, PCS is not activated there, which, apparently, is energetically favorable considering the high energy cost of PC biosynthesis [48,63].

### 4.2. Phytochelatin-Mediated Transport of Metal(loid)s into the Vacuole

As mentioned above, PC biosynthesis directly depends on the activity of glutathione biosynthesis enzymes in the cytosol and chloroplasts (Figure 1) [186]. A significant amount of PCs was found in the vacuoles of *Nicotiana rustica* [100] and *A. thaliana* [290]. Hence, it was suggested that after binding Cd ions in the cytosol, PCs can be transported into the vacuole, where, due to the more acidic pH of the vacuolar sap, these complexes can dissociate, and the peptides can be degraded by vacuolar proteases and leave the vacuole, thus acting as a shuttle mechanism for Cd transfer (Figure 1) [100,113,114,117,155,316], which, however, has not yet been directly confirmed. On the contrary, As-PC complexes entering the vacuole can remain stable and prevent re-oxidation of arsenite due to the acidic pH of the vacuole, which leads to the accumulation of high concentrations of As-PC complexes there [79,114].

Three types of Cd-PC complexes, differing in molecular weight, have been identified. The low-molecular-weight (LMW) complex [100,117] and the medium-molecular-weight (MMW) complex [159] differ in the degree of polymerization, and appear to be formed immediately after PC biosynthesis in the cytosol (Figure 1). The high-molecular-weight (HMW) complex, with the highest degree of polymerization, was isolated, for example, from *B. juncea* [317], *S. lycopersicum* [318], *Z. mays* [93], *A thaliana* [49], and *Canavalia lineata* [197]. A distinctive feature of this complex is the presence of acid-labile sulfide (S^2−^), which increases its affinity for Cd ions, the number of Cd ions bound per molecule, the stability of the complex, and its resistance against proteolytic degradation [38,115,129,155]. For example, upon the formation of an HMW Cd-PC complex in *Phaeodactylum tricornutum* cells, the Cd/SCys ratio increased from 0.6 to 1.6 [319]. It was shown that the enzymes of purine metabolism can take part in the reactions leading to the formation of S^2−^ in *S. pombe* [317,320]. It is assumed that in higher plants, S^2−^ comes from the chloroplasts [129]. The HMW complex is probably formed in the vacuole [115,321,322] or at the tonoplast level [170] and facilitates more efficient metal binding and detoxification (Figure 1).

Early experiments on tonoplast vesicles isolated from the roots of *Avena sativa* showed that Cd-PC complexes are transported by ABC (ATP-binding cassette) transporters (Figure 1) [78], one of the functions of which is to transport glutathione complexes with various secondary metabolites and xenobiotics across the tonoplast [177,323,324,325]. Two transmembrane domains (TMD) (or membrane-spanning domains, MSD) determine the substrate specificity of the transporter, and two nucleotide-binding domains (NBD) are responsible for the coupling of ATP hydrolysis and substrate transport [177,323,324,325,326]. Binding of ATP to NBD induces a conformational change in TMD, causing the substrate to enter the niche in the membrane created by the transporter. After ATP hydrolysis and phosphate release, a subsequent rearrangement of both domains occurs, which is accompanied by the release of the substrate on the other side of the membrane, as well as the release of ADP [15]. The first identified protein that carries out the ATP-dependent transport of both PCs and LMW Cd-PC complexes into the vacuole was HMT1 (heavy metal tolerance-factor 1), found in LK-100, a mutant of *S. pombe* that is not capable of forming HMW Cd-PC complexes. HMT1 belongs to the MRP (multi-drug resistance proteins) or ABCC (ATP-binding cassette subfamily C proteins) subfamily [326], is located at the tonoplast, consists of one TMD domain and one NBD domain, and is encoded by the *HMT1* gene [321,322]. Later, *HMT1* homologues were identified in *C. elegans* [327] and *Drosophila melanogaster* [328], but have not yet been found in plants. It is assumed that HMT1 has a high substrate specificity for glutathione [329], although yeast HMT1 in *A. thaliana* mutants was involved in the entry of metal complexes with PCs into the vacuoles of root cells, limiting metal translocation into the shoots [330].

In *Saccharomyces cerevisiae*, Cd is transported across the tonoplast mainly as a complex with glutathione [331,332]. The Mg-ATP-dependent transporter YCF1 (yeast cadmium factor 1) mediates the transport of Cd-GS_2_ [332,333], Hg-GS_2_ [334], and As-GS_3_ [335] into the vacuole. YCF1 belongs to the ABCC type and is encoded by the *ScYCF1* gene, the increased expression of which leads to increased Cd tolerance in transgenic *A. thaliana* plants [333]. In *A. thaliana*, the transport of Cd-PC complexes into the vacuole is carried out by the tonoplast transporters AtABCC1/2/3 (Figure 1) [304,336,337,338]. The expression of the *AtABCC3* gene is regulated by Cd, and the activity of the AtABCC3 transporter depends on metal concentration and is coordinated with the activities of AtABCC1 and AtABCC2 [338,339]. The expression of the *AtABCC1* and *AtABCC2* genes is positively regulated by transcriptional factor AtMYB40 [282]. In the *abcc3* mutant of *A. thaliana*, as well as in the double mutant *abcc1/abcc2*, Cd accumulated in the cytosol, whereas in plants overexpressing *AtABCC3*, the Cd content in the vacuole was higher than in the wild type plants [304,338]. The ABCC1/2 transporters are also involved in the transport of PC complexes with As(III) and, apparently, Zn, Cu(II), Mn, and Hg, including PheHg and other compounds, into the vacuole [57,304,340,341,342]. Furthermore, in a phosphomimetic mutant study it was shown that phosphorylation of the Ser-846 residue in the linker region between NBD1 and TMD2 regulates the activity of AtABCC1, which is necessary for As sequestration in the vacuole [342]. In yeast heterologous expression analyses, OsABCC1 enhanced PC-dependent As tolerance but did not affect Cd tolerance [341], suggesting that OsABCC1 has a high selectivity for the As-PC complex but a low affinity for the Cd-PC complex. The degree of PC polymerization can probably also affect the efficiency of the transport of their complexes with metals across the tonoplast. It is assumed that the complexes of Cd with synthetic PCs with a high degree of polymerization cannot be easily transported across the tonoplast as compared to the complexes with a low degree of polymerization [343]. Comparative analysis showed that *abcc1/abcc2* double mutants, as well as *cad1-3* and *cad1-6* mutants, which have a T-DNA insertion disrupting the C-terminal half of the Phytochelatin_C domain of AtPCS1, were hypersensitive to As(III), Hg (II), as well as to PheHg [57,221]. Therefore, both PC biosynthesis and transport of Me-PC complexes into the vacuole are important components of PC-dependent detoxification of toxic elements in plant cells (Figure 1). It is assumed that PC biosynthesis is regulated according to the principle of negative feedback: the more Me-PC complexes enter the vacuole, the more PCs are synthesized in the cytosol [340]. On the other hand, the amount of metal entering the conductive tissues and aboveground organs depends on the efficiency of metal sequestration in the vacuoles of the root cortical cells [19,341].

### 4.3. Participation of Phytochelatins in Long-Distance Transport of Metal(loid)s

Phytochelatins can take part not only in the detoxification, but also in the long-distance transport of metals. This is confirmed by numerous studies that assessed the changes in the concentration of metal(loid)s in transgenic plants or mutants. For example, an increase in the Cd concentration in transgenic *A. thaliana* plants was shown upon the expression of the *BnPCS* gene from *B. napus* [261], the *NnPCS1* gene from *N. nucifera* [267], the *AsPCS1* gene from *Allium sativum* [254], and the *ZmPCS1* gene from *Z. mays* [230]; though it was not observed when the *VsPCS1* gene from *V. sativa* was expressed in transgenic *A. thaliana* [52]. Overexpression of *CdPCS1* from *C. dactylon* [268], *AtPCS1* from *A. thaliana* [50,287], and *PtPCS* from *P. tomentosa* [277] also led to an increase in Cd accumulation in transgenic *N. tabacum* plants compared to the wild type, which, however, was not observed in transgenic tobacco lines with the overexpression of the *NtPCS1* gene [279]. The physiological reasons for these discrepancies are not clear yet. The expression of the *TaPCS1* gene in *cad1-3* transgenic mutants led not only to the restoration of PC biosynthesis, but also to an increase in the Cd root-to-shoot translocation [51]. The expression of the *CdPCS1* gene from *Ceratophyllum demersum* in transgenic *O. sativa* plants also led to an increase in PC concentration compared to non-transgenic plants, which was accompanied by an increase in As accumulation in the roots and shoots, while its concentration in the caryopses decreased [84]. At the same time, in the *OsPCS1* mutant of *O. sativa*, the As concentration in the caryopses increased, while the Cd concentration decreased, which indicates the existence of different PC-dependent pathways of As and Cd transport [240]. The analysis of *cad1-3* and *cad1-6* mutants *of A. thaliana* also suggested the existence of a PC-dependent pathway of Zn root-to-shoot translocation [54].

Direct analysis showed the presence of PCs in the xylem sap of *B. napus* and *B. juncea*, as well as in the phloem sap of *B. napus* [344,345]. However, no As–PC complexes were found in the xylem sap of *H. annuus* [47]. Due to the low pH values of the xylem sap (~5.5–6.2), the stability of metal complexes with PCs may be lower there compared to the phloem sap, where, due to the neutral pH values (~7.5), the stability of the complexes is rather high. Therefore, it can be assumed that phloem is the main conducting tissue for the long-distance transport of metal complexes with PCs and glutathione. Phloem transport plays an important role in the entry of metals into generative organs and seeds. The involvement of PCs in this process was confirmed by the high expression level of the *PCS1* gene in the phloem companion cells in *A. thaliana* [346]. Phytochelatins were also shown to be transported from shoots to roots in *A. thaliana* [347]. On this basis, it has been suggested that PCs are involved in metal transport from the shoots, as a result of which metal accumulation in the shoots decreases, and, consequently, the toxic effect on photosynthesis is diminished [125,345]. However, despite the presence of Me-PC complexes in the phloem and xylem sap, metals are mainly transported via conducting tissues as complexes with organic acids [19,30,348].

Glutathione is an important potential ligand for binding not only Cd, but also Cu and Zn, as a result of which there may be competition between these metal ions for binding to glutathione, as well as competition between PCs, glutathione, and other ligands for binding Cd [344,349,350] and other metal ions in the xylem vessels. Enhanced root-to-shoot translocation of Zn was shown for the transgenic lines of *A. thaliana* with elevated levels of glutathione [351], while mutants with reduced PC biosynthesis accumulated less Zn in the leaves compared to the wild type [54], which indicates the possible role of these compounds in the long-distance transport of not only Cd, but also Zn [18].

There are only a few studies on the mechanisms of the entry of S-containing ligands and their complexes with metals into conducting tissues. It was proposed that in *A. thaliana*, an oligopeptide transporter AtOPT6 can transport glutathione, PCs, and Cd complexes with these thiols into actively dividing cells around the phloem in sink organs [352]. It was also suggested that the loading of As(III)-PC_2_ and As(III)-GS_3_ complexes into the xylem vessels is mediated by the OsABCC7 transporter located on the plasma membrane of the xylem parenchyma cells in *O. sativa* roots [353].

### 4.4. Metal(loid) Detoxification in the Rhizosphere

In addition to the participation of PCs in metal entry into the vacuole and the long-distance transport through conductive tissues, their presence in root exudates has recently been shown. Under the treatment with As, (γ-Glu-Cys)_2_-Gly, (γ-Glu-Cys)_2_-Glu, (γ-Glu-Cys)_2_, as well as dimers linked by disulfide bridges [(PC_2_)_2_ and (PC_3_)_2_], were found in the root exudates of *Lupinus albus*. It is assumed that PCs can participate in As detoxification in the rhizosphere, limiting its entry into the roots, or that As(III)–(PC_2_)_2_ complexes can be exuded from the roots, possibly with the participation of ABC-type transporters [127].

The analysis of PC accumulation in plant tissues is an important biomarker for the presence of metals in the cytoplasm and the effectiveness of metal detoxification mechanisms in excluders, which is important for assessing the toxicity of metals [146,155,354], and is also an indicator of environmental pollution with metals [136,153,355].

## 5. Conclusions and Outlook

Having entered the cytoplasm, metal ions bind to various ligands involved in their transport and detoxification, and it is often not clear yet how the metal is transferred from the transporter to the ligand. In different plant species, various ligands can be present in the cytosol in different ratios. In addition to PCs and glutathione, an important role in metal binding is played by histidine, nicotianamine, metallothioneins, and organic acids, the affinity for which can vary significantly for different metal ions [9,19,175,356]. Therefore, there will be competition between the ligands for binding metal ions, and the amount of metal bound to one or another ligand can depend both on the strength and stability of the complexes formed, and on the amount of different ligands in the cell. In different species, the concentration of various ligands can vary significantly, which is especially evident when comparing excluders and hyperaccumulators. For example, PC concentration is low in hyperaccumulators [48,63,64,181,305,306] and, therefore, they do not play a significant role in the mechanisms of hyperaccumulation, which does not exclude a certain role of PCs in metal detoxification and maintenance of metal homeostasis. The low concentration of PCs and the high endogenous level of histidine and nicotianamine in the roots of hyperaccumulators [19] restrict metal entry into the vacuoles of root cells, facilitating their radial transport and loading into the xylem vessels. Excluders, on the contrary, have a higher level of PCs in their roots and a lower level of N-containing ligands [19], which determines the accumulation of metals in the vacuoles of root cells and their limited entry into the shoots. Obviously, the mechanisms of metal detoxification and the contribution of PCs and other metal-binding ligands to plant metal tolerance require further comparative studies.

Different metals can affect the production of low-molecular-weight ligands in cells to a different degree, and this effect may differ for different ligands, which, accordingly, will lead to a change in the buffer capacity of the cytosol [19]. There are few works that studied the combined effects of different metals on the concentration of PCs and glutathione [146,148,149,357]. However, this line of research is promising, since plants often encounter polymetallic stress in natural habitats, and PCs are considered as indicators of metal pollution [136,153,355].

Despite the extensive literature on the concentration of PCs in various plant organs summarized in this review, there is much less data on the structure of PC complexes with metals and their localization in various plant tissues. What makes it more complicated is the fact that PCs can have different degrees of polymerization and form complexes of different compositions [38,120]. The distribution and accumulation of metals can differ significantly not only in root and shoot tissues, but also in different cells of the same tissue [19,174]. Since the biosynthesis of PCs is induced by metals, it can be expected that PC concentration in plant tissues will be different. However, due to the difficulties in visualizing the ligands and their complexes with metals in plants tissues, such studies are practically absent.

The key enzyme that determines PC biosynthesis is PCS, which is present not only in the cytosol but also in various organelles [52,203,228,231,233,234], which may determine the presence of a wide range of functions. Some of them are already known [124,229,291,294,295], but we do not have a complete understanding of the role of PCS in various cell compartments. Recently, more works have appeared that testify to the biosynthesis of iso-PCs in the representatives of certain plant families. Despite the fact that significant progress has been made in deciphering the molecular mechanism of PC biosynthesis, we still know very little about the biosynthesis of iso-PCs, their functional significance, and also about the evolutionary aspects of their appearance in certain systematic groups. The information on the regulation of PC biosynthesis, including the data on hormonal regulation, is also very incomplete.

There is a certain amount of conflicting data regarding the involvement of PCs in the mechanisms of plant metal tolerance and their contribution to plant metal accumulation capacity, which is summarized in this work and in the review [214]. The resolution of the contradictions that have arisen is impossible without the elucidation of the pathways of radial and long-distance transport of metals in plants and the contribution of low-molecular-weight ligands to these processes. Recently, it has been proposed to use genetic engineering methods to create transgenic plants with enhanced metal tolerance for practical purposes [14,358,359]. However, even targeted creation of transgenic plants, for example, with overexpression of *PCS* genes, can lead to either an increase or a decrease in metal tolerance and plant ability to transport metals from roots to shoots (see above). Another interesting direction is the creation of plants with overexpression of synthetic genes encoding peptides similar to PCs and having the structure of Met(Glu-Cys)_n_Gly [343] or MetHis_6_[α-Glu(Cys)]_6_Gly [360]. It is often proposed to use transgenic plants with a high metal tolerance and the capacity to accumulate metals in aboveground organs for phytoremediation; however, this approach is also limited by the existing risks [361].

Understanding the physiological mechanisms that determine metal tolerance and the ability of plants to selectively accumulate metals in aboveground or underground organs is of fundamental and practical importance. The discovery of PCs in the representatives of various kingdoms of living organisms raises the question of their origin in the process of evolution and possible reasons for their wide distribution in different taxa. The latter may be associated not only with the high concentration of metals in the environment at the dawn of life, but also with a wider range of PC functions, which, however, requires further studies and confirmation. Since PCs play an important role in the mechanisms of metal detoxification and maintenance of metal homeostasis, their study is a promising direction for further research and they have a certain potential for the use in the development of phytoremediation, biofortification, and phytomining technologies.

## Figures and Tables

**Figure 1 ijms-24-02430-f001:**
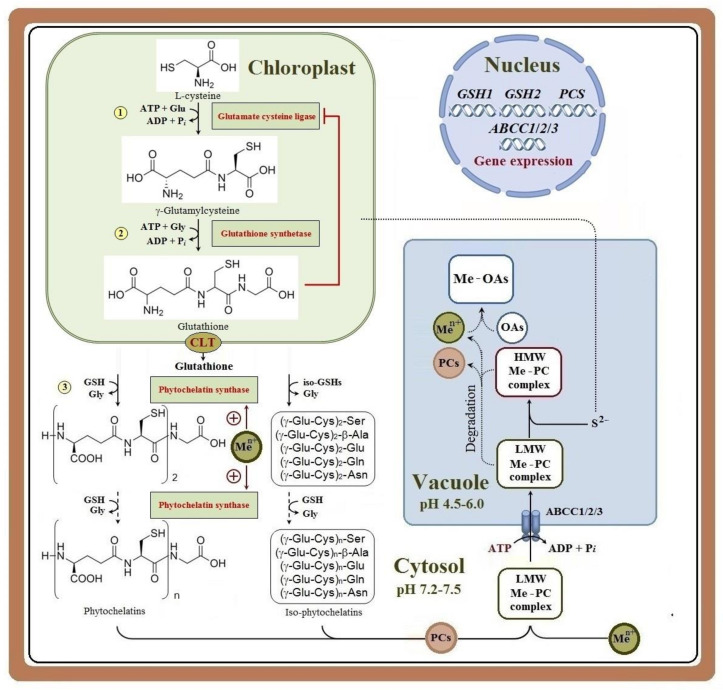
Phytochelatin-mediated pathway of metal detoxification in plants.

**Table 1 ijms-24-02430-t001:** Classification of phytochelatins.

PC Family	Peptide Structure	Identification
Phytochelatins		
PC_n_-Gly	(γGlu-Cys)_n_-Gly	PC_n_
Iso-phytochelatins		
PC_n_-Ser	(γGlu-Cys)_n_-Ser	iso-PC_n_(Ser)
PC_n_-Ala	(γGlu-Cys)_n_-Ala	iso-PC_n_(Ala)
PC_n_-βAla	(γGlu-Cys)_n_-βAla	iso-PC_n_(βAla)
PC_n_-Glu	(γGlu-Cys)_n_-Glu	iso-PC_n_(Glu)
PC_n_-Gln	(γGlu-Cys)_n_-Gln	iso-PC_n_(Gln)
PC_n_-Asn	(γGlu-Cys)_n_-Asn	iso-PC_n_(Asn)
PC_n_-Cys	(γGlu-Cys)_n_-Cys	iso-PC_n_(Cys)
des-Gly-PC_n_	(γGlu-Cys)_n_	des-Gly-PC_n_
des-γGlu-PC_n_-Gly	Cys-(γGlu-Cys)_n−1_-Gly	des-γGlu-iso-PC_n_(Gly)
des-γGlu-PC_n_-Ser	Cys-(γGlu-Cys)_n−1_-Ser	des-γGlu-iso-PC_n_(Ser)
des-Cys-PC_n_-Glu	Glu-(γGlu-Cys)_n−1_-Glu	des-Cys-iso-PC_n_(Glu)

**Table 2 ijms-24-02430-t002:** The structure of phytochelatins identified in different plant species depending on the metal(loid) concentration in the medium and the duration of exposure.

Species	Plant Material	Metal(loid) Concentration	Duration of Exposure	PC Structure	Ref.
**Amaranthaceae**					
*Amaranthus hypochondriacus*	Leaves	100 mg/kg Cd	3 months	PC_2–4_	[40]
*Pfaffia glomerata*	Roots	25, 50, 100 µM As	28 days	PC_2–4_	[41]
*Spinacia oleraceae*	Roots, leaves	1, 3, 5, 9 mg/L Cd	1, 3, 5, 7, 9, 14 days	PC_2–4_	[42]
**Apiaceae**					
*Datura innoxia*	Cell culture	250 µM Cd	up to 2 h	PC_2–5_	[43]
**Apocynaceae**					
*Rauvolfia serpentina*	Cell culture	200 µM Cd	1–9 h	PC_2–5_	[44]
**Asteraceae**					
*Dittrichia viscosa*	Roots, shoots	5, 10, 15 mg/L Cd	10 days	PC_2–4_, iso-PC_2–3_(Cys), des-γGlu-iso-PC_2–3_(Gly), des-Gly-PC_2–3_	[45]
*Eupatorium cannabinum*	Roots	11 mg/L As	20 days	PC_2–4_, des-Gly-PC_2–4_, γGlu-iso-PC_3_(Gly)	[46]
	Leaves			PC_3–4_, des-Gly-PC_2_, des-γGlu-iso-PC_2_(Gly)	
*Helianthus annuus*	Roots, stems, leaves	66 µmol/L As	1–96 h	PC_2–3_	[47]
**Brassicaceae**					
*Arabidopsis halleri*	Roots	25 µM Cd 100 µM Cd	7 days	PC_2–4_ PC_2–5_	[48]
	Shoots	25 µM Cd 100 µM Cd		n.d. * PC_2–4_	
*Arabidopsis thaliana*	Seedlings	30 µM Cd	1 day	PC_2–4_	[49]
		30, 90 µM Cd	3, 9 days	PC_2–4_	[50]
		20 µM Cd	3 days	PC_2–4_	[51]
		10 µM Cd	5 days	PC_2–4_	[52]
	Roots, leaves	5 µM Cd	1, 3, 7, 14, 21 days	PC_2–5_	[53]
	Roots, shoots	10, 25 µM Cd	7 days	PC_2–5_	[48]
	Leaves	7.5 mg/kg Cd 970 mg/kg Zn	24 days	PC_2–3_	[54]
	Roots, shoots	10 µM Cd 5 µM Cu 150 µM Zn	12 days	PC_2–5_	[55]
	Roots, shoots	20 µM Cd	72 h	PC_2–4_	[56]
	Roots, shoots	1 µM Hg, 0.1 µM phenylmercury	4 days	PC_2_	[57]
	Cell culture	200 µM Cd	1, 4, 8, 11, 24, 48 h	PC_2–4(5)_, iso-PC_3–4_(Ser), iso- PC_3_(Glu), iso-PC_3–4_(*β*Ala), iso-PC_3–4_(Gln)	[58]
		50 µM Cd	1 day	PC_2–5_, iso-PC_3–4_(*β*Ala)	
		400 µM Cd	1 day	PC_2–5_, iso-PC_3–4_(Ser), iso- PC_3_(Glu), iso-PC_3–4_(*β*Ala), iso-PC_3–4_(Gln)	
*Armoracia rusticana*	Roots	1000 µM Cd	3 days	PC_3–4_	[59]
*Brassica chinensis*	Roots	200 µM Hg	3 days	PC_2–4_	[60]
*Brassica juncea*	Roots, shoots	50, 200 µM Cd	7 days	PC_2–4_	[61]
	Roots, leaves	500, 1000, 2000 mg/kg Pb	45 days	PC_2–3_	[62]
*Brassica oleracea*	Seedlings	90 µM Cd	21 days	PC_2–6_	[44]
*Noccaea caerulescens* *(Thlaspi caerulescens)*	Roots, shoots	1–50 µM Cd	4 days	PC_2–3_	[63]
	Roots shoots	25, 100 µM Cd 25, 100 µM Cd	7 days 7 days	PC_2–4_ n.d.	[48]
	Roots, shoots	5–500 µM Cd	14 days	PC_2–4_	[64]
*Sinapis alba*	Leaves	0.5, 1 mg/L Pd	2 weeks	PC_2–4_	[65]
*Thlaspi arvense*	Roots, shoots	1–50 µM Cd	4 days	PC_2–3_	[63]
**Caryophyllaceae**					
*Silene vulgaris*	Cell culture	20 µM Cd	3 days	PC_2–4_	[44]
	Roots	0.3, 1, 45, 135, 180 µM Cd	3 days	PC_2–4_	[66]
		40 mmol m^−3^ Cd	21 days	PC_2–3_	[67]
**Crassulaceae**					
*Sedum alfredii*	Shoots	500 µM Cd	8 days	PC_2–4_	[68]
**Cucurbitaceae**					
*Cucumis sativus*	Roots	10–250 µM Sb	28 days	PC_2, 3_	[69]
**Fabaceae**					
*Arachis hypogaea*	Roots	10 µM Cd	30 days	PC_2–4_	[70]
*Glycine max*	Roots	20 µM Cd	4 days	iso-PC_2–7_(*β*Ala)	[71]
*Pisum sativum*	Roots	20 µM Cd	3 days	PC_2–3_, iso-PC_3_(*β*Ala)	[72]
		1–120 µM Cd	1–9 days	PC_2–4_, iso-PC_2–4_(*β*Ala)	[73]
**Lamiaceae**					
*Clinopodium vulgare*	Roots	20 mg/kg Cd 250 mg/kg Pb 400 mg/kg Cu 400 mg/kg Zn	95, 105 days	All metals: PC_2–5_, iso-PC_2_(Ala); Cu: iso-PC_2_(Glu), iso-PC_2–4_(Cys); Cu, Cd, Pb: des-γGlu-iso-PC_2–3_(Gly); Cu, Cd, Pb: des-Gly-PC_2_	[74]
	Shoots	20 mg/kg Cd 250 mg/kg Pb 400 mg/kg Cu 400 mg/kg Zn	95, 105 days	All metals: PC_2–5_Cu, Cd, Pb: des-γGlu-iso-PC_2–3_(Gly) Cu, Cd, Pb: des-Gly-PC_2_	
*Perilla frutescens*	Roots, stems, leaves	2, 5, 10 mg/L Cd	14 days	PC_2–4_	[75]
			21 days	PC_2–3_	
**Marchantiaceae**					
*Marchantia polymorpha*	Gametophyte	10–36 µM Cd	6–120 h	PC_2–4_	[76]
		10, 20, 36 µM Cd	72 h	PC_2–4_	[77]
**Poaceae**					
*Agrostis tenuis*	Cell culture	20 µM Cd	3 days	PC_2–4_	[44]
*Avena sativa*	Roots	10 µM Cd	4 days	PC_2–3_	[78]
*Holcus lanatus*	Roots	5, 15, 40, 800 µM As	7 days	PC_2–4_	[79]
*Lolium perenne*	Roots, shoots	20, 80 µM Cd	9 days	PC_2–6_	[80]
		20, 80 µM Cd	216 h	PC_2–6_	[81]
*Oryza sativa*	Roots	10 µM Cd 20 µM As	7 days	PC_2–3_, iso-PC_2_(Ser), iso-PC(Glu)	[82]
	Roots, stems, leaves	50, 100 µM Cd	7 days	PC_2–4_, iso-PC_2–3_(Gln), iso-PC_2–3_(Asn), iso-PC_2_(Cys), des-γGlu-iso-PC_3_(Ser), des-Cys-iso-PC_2_(Glu), des-Gly-PC_2–4_, iso-PC_2–4_(Ser), iso-PC_2–4_(Glu)	[83]
	Roots, shoots	100 µM As	10 days	PC_2–4_	[84]
	Roots, shoots	50 µM Cd	14 days	PC_2–4_	[85]
*Panicum maximum*	Roots	100 µM Cd	9 days	PC_2–4_, des-Gly-PC_2–4_, iso-PC_2–3_(*β*Ala), des-γGlu-iso-PC_2_(Gly)	[86]
	Stems			PC_3, 5, 6_, iso-PC_2–4_(*β*Ala), des-Gly-PC_4,_ des-γGlu-iso-PC_2–3_(Gly)	
	Leaves			PC_6_, iso-PC_4_(*β*Ala)	
*Phragmites australis*	Roots	100 µM Cd	21 days	PC_2–4_	[87]
*Secale cereale*	Roots	50, 250 µM Cd	3, 6, 12, 24 h, 3, 7, 14 days	PC_2–3_, iso-PC_2–3_(Ser), des-Gly-PC_2–3_	[88]
	Shoots	50 µM Cd 250 µM Cd	7,14 days 7 days	PC_2–3_, iso-PC_2–3_(Ser), des-Gly-PC_2–3_	
*Triticum aestivum*	Roots	1 µM Cd	12 days	PC_2–3_, iso-PC_2–3_(Ser)	[89]
		30 µM Cd		PC_2–4_, iso-PC_2–4_(Ser)	
	Roots	1 mM Cd	10, 20 days	PC_2–4_, des-Gly-PC_2–3_	[90]
	Shoots			PC_2–4_	
*Triticum turgidum var. durum*	Roots	1 µM Cd	12 days	PC_2–3_, iso-PC_2–3_(Ser)	[89]
		30 µM Cd		PC_2–4_, iso-PC_2–4_(Ser)	
*Triticum vulgare*	Roots	50, 250 µM Cd	3, 6, 12, 24 h, 3, 7, 14 days	PC_2–3_, iso-PC_2–3_(Ser), des-Gly-PC_2–3_	[88]
	Shoots		7,14 days		
*Zea mays*	Roots	50 µM Cd	3, 6, 12, 24 h, 3, 7, 14 days	PC_2–4_, iso-PC_2_(Glu), des-Gly-PC_2–3_	[88]
		0.01 µM Cd 0.05, 0.1 µM Cd 0.5, 1, 10 µM Cd 3 µM Cd	1 day 1 day 1 day 2, 4, 6 h; 1, 2 days	PC_2_ PC_2–3_ PC_2–4_ PC_2–4_	[91]
		3 µM Cd	1–7 days	PC_2–4_, des-Gly-PC_2–4_, iso-PC_2–4_(Glu)	[92,93]
	Roots	38 µM Cd	5 days	PC_2–5_, iso-PC_2–5_(Ser)	[94]
	Shoots			PC_2–3_, iso-PC_2–3_(Ser)	
	Roots	10, 15, 25 µM Cd	14 days	PC_2–3_, PC_7, 8_, PC_10_, PC_4, 9_ (10 and 15 µM Cd), PC_5_ (25 µM Cd), PC_6_ (10 and 25 µM Cd)	[95]
	Leaves			PC_2,3,6,8,10_ PC_4_ (25 µM Cd)	
	Seedlings	20 µM Cd	3 days	PC_2–4_	[44]
**Pontederiaceae**					
*Pontederia crassipes (Eichhornia crassipes)*	Seedlings	20 µM Cd	3 days	PC_2–4_	[44]
	Roots	1, 2.5, 3.5 ppm Cd	45 days	PC_2–4_	[96]
**Proteaceae**					
*Banksia seminuda*	Roots	10–250 µM Sb	120 days	PC_2, 3_	[69]
*Hakea prostrata*	Roots	10–250 µM Sb	120 days	PC_2, 3_	[69]
**Pteridaceae**					
*Pteris cretica*	Fronds	100 mg/kg As	1 year	PC_2_	[97]
**Rubiaceae**					
*Rubia tinctorum*	Cell culture	100 µM Cd	3 days	PC_2–4_, des-Gly-PC_2–4_	[98]
**Salicaceae**					
*Salix atrocinerea*	Roots	18 mg/L As	1, 3, 10, 30 days	PC_2–3_, des-Gly-PC_3_, des-γGlu-iso-PC_2–3_(Gly)	[99]
	Leaves			des-Gly-PC_2–4_	
**Solanaceae**					
*Nicotiana rustica*	Leaves	20 µM Cd	7 days	PC_3–4_	[100]
*Nicotiana tabacum*	Cell culture	250 µM Cd	3 days	PC_4–5_	[101]
	Seedlings	30, 90 µM Cd	3, 9 days	PC_2–4_	[50]
*Solanum lycopersicum* (*Lycopersicon esculentum*)	Cell culture	100 µM Cd	2 h	PC_3–5_	[102]
		100, 300 µM Cd	1–12 days	PC_2_	[103]
		600 µM Cd	36 h	PC_3–4_	[104]
		10, 50, 100 µM Cd	4 days	PC_2–4_	[105]
		100 µM Cd	7 days	PC_2–4_	[106]
	Roots	3 µM Cd	7 days	PC_3_	[107]
	Seedlings	90 µM Cd	21 days	PC_2–6_	[44]
	Roots, leaves	25, 100 µM Cd	14 days	PC_2–4_	[108]
**Vitaceae**					
*Vitis vinifera*	Roots	100 mg/L Hg	3 days	PC_2–4_	[109]

* n.d.—not determined.

**Table 3 ijms-24-02430-t003:** Phytochelatin synthase genes identified in plant species from different families.

Family	Species	Genes	References
**Bryophytes**			
Marchantiaceae	*Marchantia polymorpha*	*MpPCS*	[33]
**Pteridophytes**			
Pteridaceae	*Pteris vittata*	*PvPCS1*	[253]
**Angiosperms**			
Alliaceae	*Allium sativum*	*AsPCS1*	[254]
Amaranthaceae	*Salicornia europaea*	*SePCS1*	[141]
Arecaceae	*Phoenix dactylifera*	*PdPCS1*	[255]
Asteraceae	*Helianthus annuus*	*HaPCS*	[256]
	*Lactuca sativa*	*LsPCS1, LsPCS2*	[257,258]
	*Tagetes patula*	*TpPCS1*	[259]
Brassicaceae	*Arabidopsis halleri*	*AhPCS1, AhPCS2*	[48]
	*Arabidopsis thaliana*	*AtPCS1, AtPCS2*	[215,225,244,252,260]
	*Brassica juncea*	*BjPCS1*	[173]
	*Brassica napus*	*BnPCS*	[261]
	*Brassica rapa*	*BrPCS1, BrPCS2*	[133]
	*Noccaea caerulescens*	*NcPCS1, NcPCS2*	[48]
	*Noccaea japonicum*	*NjPCS*	[262]
Ceratophyllaceae	*Ceratophyllum demersum*	*CdPCS1*	[84]
Chenopodiaceae	*Suaeda salsa*	*SsPCS*	[263]
Convolvulaceae	*Ipomoea pes-caprae*	*IpPCS1*	[226]
Fabaceae	*Cajanus cajan*	*CcPCS1*	[182]
	*Lotus japonicus*	*LjPCS1*, *LjPCS2, LjPCS3*	[264]
	*Medicago sativa*	*MsPCS1, MsPCS2*	[135,265,266]
	*Vicia sativa*	*VsPCS1*	[52]
Moraceae	*Morus notabilis*	*MnPCS1, MnPCS2*	[243]
Nelumbonaceae	*Nelumbo nucifera*	*NnPCS1*	[267]
Poaceae	*Arundo donax*	*AdPCS1*, *AdPCS2, AdPCS3*	[213]
	*Cynodon dactylon*	*CdPCS1*	[268]
	*Orysa sativa*	*OsPCS1*, *OsPCS2 etc.*	[82,233,269,270,271]
	*Panicum maximum*	*PmPCS2*	[86]
	*Paspalum vaginatum*	*PvPCS1*, *PvPCS2*	[272]
	*Phragmites australis*	*PaPCS*	[273]
	*Saccharum officinarum*	*SoPCS*	[232,274]
	*Triticum aestivum*	*TaPCS1*	[248,275]
	*Zea mays*	*ZmPCS1*	[230]
Rosaceae	*Malus hupehensis*	*MhPCS*	[276]
Salicaceae	*Populus tomentosa*	*PtPCS*	[277]
	*Populus trichocarpa*	*PtPCS1*	[278]
Solanaceae	*Nicotiana tabacum*	*NtPCS1*	[279]
	*Solanum lycopersicum*	*SlPCS*	[108,280]
Urticaceae	*Boehmeria nivea*	*BnPCS1*	[203]

## Data Availability

No new data were created or analyzed in this study. Data sharing is not applicable to this article.

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
