# Peer review of "Phytochelatins: Sulfur-Containing Metal(loid)-Chelating Ligands in Plants"

_ijms, 2023, doi:10.3390/ijms24032430_

Round 1
Reviewer 1 Report
This manuscript exposes a literature review and therefore does not present data generated or quantitatively analysed by the authors in the manuscript. Thus, the manuscript's merit relies upon the impressive amount of reviewed and consistently compiled literature. On the other hand, a limitation of the presented discussion is that it only involves the presence of phytochelatins and not quantifying the dynamic of their accumulation in different species under different conditions. However, I understand that overcoming this limitation would escape this review's objective. Nevertheless, the manuscript will undoubtedly be a valuable reference in its concentration area.
Author Response
We thank the reviewer for the valuable comments and suggestions. We have added a paragraph discussing the dynamics of PC accumulation (Lines 101-114). Unfortunately, most of the studies of different species were performed in different conditions (duration of exposure, metal concentration, soil/hydroponics, climate room conditions etc.), which complicates direct comparison of the data obtained.
Reviewer 2 Report
This manuscript reviews a comprehensive review of biosynthesis and regulation, structure and accumulation, transport and physiological role of Phytochelatins in the vacuole of plant cell. This review is of great importance to study the phytochelatins for the development of technologies used in phytoremediation, biofortification and phytomining. In my opinion, this manuscript is well-written and provides a thoroughly new insight for the further research of the phytochelations in plant. In the Line 59-61, there are two Cd symbols in this sentence. Is the other Cr? And there some abbreviations need to be carefully checked and noted such as CA in line 347. I recommend this manuscript to be accepted in this present form after the above-mentioned is corrected.
Author Response
We thank the reviewer for the valuable comments and suggestions. Yes, in lines 59-61 there are indeed two 'Cd' symbols, since Noccaea caerulescens is a hyperaccumulator of Zn, Cd and Ni, whereas Arabidopsis halleri is a hyperaccumulator of Zn, Cd and Pb. We have checked the abbreviations. We have underlined 'c' in the text 'clan CA of the papain-like cysteine ​​proteases' (now line 357), as each clan of proteases (peptidases) is identified with two letters, the first one representing the catalytic type of the families included in the clan (https://www.ebi.ac.uk/merops/cgi-bin/clan_index?type=P; Rawlings et al., 2006; Shindo et al., 2008). Thus, for cysteine protease the clan name starts with C.